# Current Evidence and Perspectives of Cluster of Differentiation 44 in the Liver’s Physiology and Pathology

**DOI:** 10.3390/ijms25094749

**Published:** 2024-04-26

**Authors:** Jinsol Han, Chanbin Lee, Youngmi Jung

**Affiliations:** 1Department of Integrated Biological Science, College of Natural Science, Pusan National University, Pusan 46241, Republic of Korea; wlsthf1408@pusan.ac.kr; 2Institute of Systems Biology, College of Natural Science, Pusan National University, Pusan 46241, Republic of Korea; chanbin@pusan.ac.kr; 3Department of Biological Sciences, College of Natural Science, Pusan National University, Pusan 46241, Republic of Korea

**Keywords:** cluster of differentiation 44, liver disease, metabolism, inflammation, fibrosis

## Abstract

Cluster of differentiation 44 (CD44), a multi-functional cell surface receptor, has several variants and is ubiquitously expressed in various cells and tissues. CD44 is well known for its function in cell adhesion and is also involved in diverse cellular responses, such as proliferation, migration, differentiation, and activation. To date, CD44 has been extensively studied in the field of cancer biology and has been proposed as a marker for cancer stem cells. Recently, growing evidence suggests that CD44 is also relevant in non-cancer diseases. In liver disease, it has been shown that CD44 expression is significantly elevated and associated with pathogenesis by impacting cellular responses, such as metabolism, proliferation, differentiation, and activation, in different cells. However, the mechanisms underlying CD44’s function in liver diseases other than liver cancer are still poorly understood. Hence, to help to expand our knowledge of the role of CD44 in liver disease and highlight the need for further research, this review provides evidence of CD44’s effects on liver physiology and its involvement in the pathogenesis of liver disease, excluding cancer. In addition, we discuss the potential role of CD44 as a key regulator of cell physiology.

## 1. Introduction

The liver plays vital roles in nutrient and energy metabolism [1,2,3]. It is composed of various types of cells, including hepatocytes, liver sinusoidal endothelial cells (LSECs), resident macrophage Kupffer cells (KCs), hepatic stellate cells (HSCs), and biliary epithelial cells (BECs) [4,5,6]. Each type of cell has a specific function, and they closely communicate with each other under both healthy and injured conditions [7,8,9]. In a healthy liver, hepatocytes are in a quiescent state [10,11]. However, liver damage leads to massive hepatocyte death [12,13,14]. The dying hepatocytes release cytokines and damage-associated molecular patterns (DAMPs), which trigger inflammation and fibrosis. During this process, pathological interactions between cells take place, and these interactions are mediated by numerous ligand–receptor signaling pathways [12,13,14,15,16,17]. Among these pathways, cluster of differentiation 44 (CD44) is emerging as a critical regulator of cellular mechanisms, especially in the liver.

CD44, a cell surface glycoprotein, is widely expressed on different types of cells and tissues, playing essential roles in cellular physiological activities such as cell proliferation, adhesion, and migration [18,19,20,21]. Defects in CD44 can lead to pathogenic phenotypes [18]. CD44 has been studied in numerous diseases, including cancer, infections, lung disease, vascular disease, and liver disease [18,22,23,24]. Particularly, in several liver diseases, a significant increase in CD44 in patients has been reported [22,23,24]. Although the involvement of CD44 in liver disease has been suggested, most research has focused on its role in cancer, while its impact on non-cancerous liver conditions, such as metabolism, fibrosis, and inflammation, is often underestimated. Therefore, this review aims to explore the function of CD44 in the pathophysiology of the liver after providing general information on CD44 to enhance understanding of its role in the liver and suggest the need for further research on CD44 in this context.

## 2. General Information on CD44

CD44 is a transmembrane glycoprotein that belongs to the class of cell adhesion molecules [25,26,27]. It is broadly expressed in various types of cells, including endothelial cells, epithelial cells, fibroblasts, keratinocytes, and leukocytes [19,20,21,25,26,27,28,29]. Different CD44 variants are expressed depending on the cell type, which are produced by extensive alternative splicing of variable exons and different combinational insertions [19,20,21,29]. The human CD44 gene contains 19 exons and undergoes alternative splicing from exons 6 to 15, resulting in multiple CD44 variants [19,20,21,30]. These variants have several constant and common functional domains, including the N-terminal extracellular domain, stem region, transmembrane domain, and intracellular cytoplasmic domain (ICD). The N-terminal extracellular domain, which senses diverse stimuli in the external microenvironment, contains docking sites for various ligands, such as hyaluronan (HA), osteopontin (OPN), extracellular matrix (ECM) glycoproteins, growth factors, cytokines, and matrix metalloproteinases [19,20,21,26,27,28,30]. The stem region, with an insertion site of variant exon products generated by alternative splicing, is a highly variable stalk-like structure that contributes to CD44 variations. The transmembrane domain anchors CD44 in the plasma membrane [19,20,21,26,27,28,30]. The CD44ICD at the C-terminus has binding motifs for intracellular cytoskeleton proteins and serves as an important mediator of intracellular signal transduction [19,20,21,25,26,27,28,30]. It binds to cytoskeletal elements such as ankyrin, ezrin, radixin, and moesin (ERM), adjusting cytoskeletal arrangement. CD44ICD also binds to signaling molecules, including Src family kinases and activators of small Rho GTPases, resulting in cytoskeletal arrangement for cell migration [25,31,32,33]. Therefore, CD44 reacts to various extracellular signals depending on the reacting part of CD44 in different cells and tissues.

Binding of CD44 with several ligands impacts diverse cellular processes, such as proliferation, adhesion, migration, and cell-to-cell communication [19,20,21,25,26,27,28,30,31,32,33]. Among these ligands, HA and OPN are considered classical ligands of CD44 [33,34]. Upon HA binding, CD44 acts as a switch for mitosis, but its effect on mitosis varies depending on the molecular weight (MW) of the bound HA [21]. Binding of high-MW HA suppresses the Ras–cyclin D1 interaction, leading to cell-cycle arrest and inhibition of mitosis [21,35]. On the other hand, CD44 interacting with low-MW HA promotes mitosis mediated by Ras–cyclin D1 [21,36]. The CD44–HA complex recruits intracellular signaling molecules to the plasma membrane, stimulates intracellular Ca^2+^ release, and activates RhoA with Rho-binding kinase (ROK) signaling, leading to cell migration [21,37,38]. The complex also activates ankyrin and ERM proteins, which bind with actin filaments and regulate actin polymerization to rearrange the cytoskeleton, leading to cell migration [20,21,25,26]. In addition, OPN, being widely expressed by immune cells, was shown to bind to CD44 and form a complex with αVβ3 integrin [39,40]. This complex regulated macrophages’ inflammatory responses through PI3 kinase/Akt-dependent NF-κB activation, preventing pathogen infection and the recruitment and activation of inflammatory cells during the immune response.

The proteolytic cleavage of CD44 has been reported to trigger signals in cells and has recently emerged as a key mechanism in understanding CD44-regulated cellular function [21,25,41]. The interaction of CD44 with various molecules controls CD44 cleavage. The CD44 cleavage process occurs in two steps [25,41]: First, the extracellular domain of CD44 is cleaved from the membrane. Various stimuli, including ligand binding, Ca^2+^ influx, and protein kinase C activation, activate metalloproteinases such as metalloproteinase domain-containing proteins 10 and 17 (ADAM-10 and ADAM-17) and membrane type 1-matrix metalloproteinase (MT1–MMP) [41,42,43,44,45]. These activated metalloproteinases cut the CD44 stem region, which contains proteolytic cleavage sites, releasing the CD44 extracellular domain [41,42,43,44]. The second step includes intramembranous cleavage by presenilin-dependent ɣ-secretase, producing a 25 kDa CD44ICD. The soluble extracellular domain of CD44 (sCD44), generated by CD44 cleavage, causes the detachment of CD44-expressing cells from the ECM and promotes cell migration [45]. The CD44ICD is released into the cytoplasm and, with its nuclear localization signal, translocates into the nucleus, where it acts as a transcriptional regulator [25]. CD44ICD in the nucleus cooperates with the transcriptional coactivator CBP/P300 to initiate CD44 transcription, leading to an increase in CD44 itself. CD44ICD also increases cyclin D expression and promotes cell-cycle progression [25,46]. In addition, CD44ICD enhances cellular hypoxic responses including autophagy, angiogenesis, and apoptosis by regulating hypoxia-related genes, such as hypoxia-inducible factor 2a, max interactor 1, bcl-2 interacting protein 3, adrenomedullin, basic helix–loop–helix family member E40, and N-myc downstream regulated 1 [47].

## 3. CD44 in Liver Disease

Hepatic expression of CD44 is significantly altered in various pathological conditions [22,23,24,48]. CD44 upregulation has been reported in liver diseases such as metabolic dysfunction-associated steatotic liver disease (MASLD), liver fibrosis, and hepatitis C virus (HCV) infection [23,24,48]. Increased CD44 at both the RNA and protein levels has been found to be associated with hepatocyte ballooning and alanine aminotransferase levels in patients with hepatic steatosis, metabolic dysfunction-associated steatohepatitis (MASH), or hepatitis B virus (HBV) infection [23,24,48,49]. In addition to intact CD44, soluble CD44 is also significantly elevated in patients with severe acute or chronic liver disease compared with healthy controls [22,23]. The increased expression and cleavage of CD44 can disrupt metabolic homeostasis and exacerbate inflammation and fibrosis in liver disease (Figure 1). Hence, this review focuses on the role of CD44 in liver pathology to provide a basic understanding of its involvement in liver disease and suggests the need for further research on CD44 in liver disease, beyond its role in cancer.

### 3.1. CD44 Is Involved in Liver Metabolism

The development of liver disease is strongly associated with the disruption of metabolic homeostasis, and there is growing evidence suggesting that CD44 is involved in the regulation of glucose and lipid metabolism in the liver [50,51]. High-fat diet (HFD)-fed obese mice were shown to have higher hepatic levels of CD44 compared to chow-fed mice [52]. CD44 expression in obese patients was significantly elevated compared with lean healthy individuals, and its expression was significantly correlated with the grade of hepatic steatosis [49]. The amount of sCD44 in serum was also higher in obese patients and mice with severe steatosis [48,53]. In line with the increase in sCD44, the hepatic expression of ADAM-10 cleaving the extracellular domain of CD44 increased in parallel with the degree of steatosis [48]. CD44 has been reported to modulate genes involved in lipid metabolism and promote lipid accumulation in the livers of HFD-treated mice [52]. In CD44-deficient mice fed an HFD, the expression of genes involved in lipid uptake and de novo lipogenesis was alleviated. Levels of the cell death-inducing DFF45-like effector (CIDE) genes, Cidea and Cidec, which play a critical role in lipid droplet formation, were dramatically downregulated by CD44 deletion. Hepatic amounts of CD36, a major fatty acid uptake protein, and several genes regulating fatty acid biosynthesis, including fatty acid synthase, Elovl-5, and Elovl-7, were also significantly reduced in CD44-knockout (KO) mice compared to wild-type (WT) mice during HFD feeding. The altered expression of these genes collectively attenuated liver steatosis in HFD-treated CD44-KO mice [48,52].

Several studies have demonstrated an association between CD44 and glucose metabolism. Kang et al. [52] showed that CD44-KO mice exhibited greater glucose tolerance and insulin sensitivity compared to WT mice when fed a high-fat diet. Neutralization of CD44 using a monoclonal antibody in obese mice resulted in a reduction in fasting blood glucose levels and insulin resistance, similar to the effects of diabetes medications such as metformin and pioglitazone [52,54]. These findings indicate that CD44 influences glucose and lipid metabolism in the liver, but the exact mechanism is poorly understood due to limited research. Therefore, further investigations are needed to fully understand the pathophysiological role of CD44 in hepatic metabolism.

### 3.2. CD44 Regulates Inflammation

CD44, an adhesion protein, is constitutively expressed on various types of inflammatory cells, such as B cells, granulocytes, monocytes, and erythrocytes. In addition, its expression is upregulated during inflammation [55,56]. Patouraux et al. [48] demonstrated a significant increase in hepatic CD44 levels in mice with steatohepatitis, caused by a methionine–choline-deficient (MCD) diet, with CD44 levels positively correlating with inflammatory markers including tumor necrosis factor-α (TNF-α), interleukin (IL)-1β, monocyte chemoattractant protein-1 (MCP-1), and C-C chemokine receptor 2. CD44-KO mice had less liver injury including inflammation and fibrosis, compared with WT mice during MCD feeding. CD44 is known to promote immune infiltration [55,57,58,59]. For infiltration of inflammatory cells into injured sites, a series of discrete steps, such as recognition of chemoattractants, tethering, rolling, adhesion, diapedesis, and migration, are required [60]. CD44 is closely associated with the chemotactic migration of inflammatory cells [57,58,59]. Macrophages deficient in CD44 showed reduced chemotaxis induced by various chemoattractants, including MCP-1 and OPN [61]. CD44 is linked to Rho-A, which induces actomyosin contractility in response to N-formyl-met-leu-phe, a leukocyte chemotactic peptide factor [62]. CD44 binding with HA activates Rho-A, leading to actomyosin contractility through ROK phosphorylation of cytoskeletal proteins. Leukocytes lacking CD44 failed to activate Rho-A for cell migration. In addition, CD44 impacts the tethering and rolling of leukocytes for infiltration into injured areas. Inflammatory cytokines stimulate HA expression on vascular endothelial cells, with HA binding to CD44 on leukocytes, facilitating leukocyte adhesion [63]. Then, CD44-mediated activation of α4β1 integrin results in high-affinity binding with VCAM-1 on endothelial cells, stabilizing leukocyte adhesion and promoting extravasation to inflamed sites.

CD44 is also involved in the activation and recruitment of T cells. CD44 expression is upregulated during T-cell activation, and CD44-positive activated T cells are highly abundant in the livers of mice injured by hepatitis virus infection, HFD, or ischemia/reperfusion, and are correlated with disease progression [64,65,66,67,68]. CD44 mediates conjugation between T cells and antigen-presenting dendritic cells (DCs) to promote T-cell activation [69,70,71]. Funken et al. [67] reported that blocking CD44 with a monoclonal antibody (mAb) interrupted interactions between DCs and T cells and induced T-cell apoptosis, inhibiting immune response in the post-ischemic liver. CD44 interacts with hyaluronan expressed on liver sinusoidal endothelial cells and leads to T-cell recruitment into the inflamed liver sinusoids [59,72]. Treatment with an anti-CD44 mAb decreased T-cell accumulation in the injured liver and reduced T-cell motility [67,73]. CD44 on platelets is reported to be implicated in recruitment of T cells in the HBV-infected liver [66]. Circulating T cells are arrested by docking to CD44 expressed in platelets, and these T cells crawl along the liver sinusoids to search for hepatocellular antigens.

Recent studies suggest that CD44 regulates the properties and activities of macrophages. CD44 suppression enhances the polarization of hepatic macrophages into anti-inflammatory M2 macrophages [48]. When exposed to DAMPs from damaged fatty hepatocytes, CD44-deleted macrophages inhibit the expression of pro-inflammatory cytokines, including TNF-α and IL-6. In addition, levels of sCD44 in plasma have been reported to be low in immunodeficiency but high in conditions of immune activation and inflammation [56]. However, the inducers causing the cleavage of the CD44 extracellular domain to release sCD44 in inflammatory responses are not yet known. Accumulating evidence suggests a positive correlation between sCD44 and various diseases, including periodontitis, colorectal cancer, head and neck cancer, and gastric cancer [57,74,75,76]. These studies indicate that sCD44 could be a promising biomarker for disease diagnosis and monitoring.

Taken together, elevated CD44 expression and its cleavage process seem to be closely related to inflammation. However, detailed mechanisms explaining CD44’s action in inflammation remain limited. Therefore, further studies are needed to gain more insight into potential anti-inflammatory strategies for targeting CD44 in liver disease.

### 3.3. CD44 Promotes Liver Fibrosis

In the damaged liver, the inflammatory response leads to fibrosis [77,78,79,80]. HSCs are the main fibrogenic cells and are also known as one of the major cell types expressing CD44 in the liver [23,80,81,82,83,84]. CD44 expression increases with HSC activation [23]. Both quiescent HSCs and activated HSCs express multiple CD44 variants [82,83]. However, activated HSCs, both in vivo and in vitro, have significantly higher expression of CD44 than quiescent HSCs. Treatment with a CD44 antibody suppressed HSC migration [83]. In various animal models of liver fibrosis, such as the MASH model and carbon tetrachloride (CCl_4_)-induced or bile duct ligation-induced liver fibrosis model, higher levels of CD44 have been reported [23,48,83,85,86]. The increased level of CD44 in CCl_4_-treated rats was much higher in deceased rats with cirrhosis than in surviving rats with cirrhosis, indicating that CD44 levels could be an indicator of overall survival in a cirrhotic model [23]. In a chronic liver congestion model generated by partial inferior vena cava ligation, upregulation of CD44 was specifically localized in HSCs. During the progression of liver fibrosis, enhanced expression of CD44 in HSCs was accompanied by an elevation of S100A4, promoting HSC activation and fibrosis. CD44 has been reported to activate the Wnt/β-catenin pathways, and β-catenin directly binds to the S100A4 promoter to activate its expression. Based on these findings, CD44 seems to contribute to liver fibrosis through a β-catenin-related pathway regulating S100A4 expression in congestive hepatopathy. Ligand binding to CD44 also promotes fibrogenic, proliferative, and invasive phenotypes of HSCs. In HSCs, the interaction of low-MW HA with CD44 promotes HSC activation by upregulating profibrotic genes, metallopeptidase inhibitor 1, and pro-inflammatory genes such as Mcp-1 and C-X-C motif chemokine ligand 1 [87]. In addition, proteolytic cleavage of CD44 has been reported to occur during HSC activation [23,88]. The serum levels of sCD44 are elevated in HCV-infected patients with fibrosis [23]. CD44ICD has also been shown to influence HSC activation [23,88]. Released CD44ICD moved to the nucleus and bound to a putative CD44ICD response element on the promoter/enhancer region of Notch1 to induce its transcription [88]. Upregulated Notch1 promoted the activation and ECM production in HSCs. Notch, as a receptor, interacts with its ligand Jagged-1, expressed on nearby liver-resident KCs and/or HSCs, and propagates the activation of profibrogenic Notch1 signaling, contributing to the exacerbation of hepatic fibrosis [88,89].

BECs are also a type of non-parenchymal cells that express CD44 [85]. In the choline-deficient, methionine-lowered, L-amino-acid-defined diet-induced MASLD model, cytokeratin 19-positive BECs have been found to express CD44, and these double-positive cells were localized in the fibrotic areas [86]. He et al. [85] also demonstrated that BECs expressing CD44 accumulated in the fibrotic areas of the portal triads, and that HA enhanced BEC proliferation. CD44–HA interaction, which enhances biliary proliferation, may play an important role in the pathogenesis of hepatic cholestasis. However, it is not yet known how CD44 expression in BECs increases, or how BECs with upregulated CD44 promote liver fibrosis in damaged livers. Recent studies suggest a crucial role for CD44 in the pathogenic changes of HSCs and BECs in injured livers. Hence, further research is required to unveil the molecular mechanisms that explain how CD44-mediated cell transitions induce liver fibrosis.

## 4. CD44 Influences Cell Proliferation and Differentiation in the Liver

Mature hepatocytes are known to barely express CD44 and its isoforms [29,90]. However, emerging evidence suggests that CD44 is specifically expressed on the surface of small hepatocytes [91,92]. CD44 expression on small hepatocytes isolated from rats rapidly decreased during maturation into mature hepatocytes, disappearing along with the increase in C/EBPα, a well-known marker of mature hepatocytes [92]. Restricted expression of CD44 on the cell membrane has been detected in human hepatocyte progenitors, but not in mature hepatocytes [92,93]. Also, CD44-positive small hepatocytes showed high growth activity and were successfully engrafted and proliferated in the livers of rats treated with retrorsine/partial hepatectomy (PHx), where liver regeneration was suppressed [91,94]. These findings suggest that CD44 can serve as a marker for small hepatocytes. Turner et al. [95] reported that human fetal liver-isolated hepatic progenitor cells (HPCs) expressed CD44 and had the potential to differentiate into both hepatocytes and BECs. When CD44-positive HPCs were cultured in HA hydrogels, their viability and proliferative phenotype were maintained for more than 4 weeks [96], and these cells did not differentiate into either biliary or hepatocyte cells, whereas HPCs cultured on plastic without HA hydrogels exhibited growth arrest and differentiated into more mature cells. These findings suggest that the interaction of CD44 with HA helps HPCs maintain their differentiation capability.

A study has reported that not only small hepatocytes but also mature hepatocytes express CD44 when the liver is partially removed and mitogenic stimuli are produced [97,98]. The liver exhibits a low rate of cellular turnover, and hepatocytes are mostly found in a quiescent state (G0) of the cell cycle under normal/healthy conditions [99,100,101]. When the liver is injured or partially removed, hepatocytes exit G0 in large numbers and undergo multiple rounds of division simultaneously, showing a remarkable capacity for regeneration. In mice with PHx livers, CD44 expression was significantly elevated during liver regeneration. Particularly, among the hepatocytes isolated from PHx livers, hepatocytes isolated 10 days after PHx had the highest CD44 expression [98]. This research showed that CD44 interacted with the glutamate–cystine transporter xCT to stabilize it and promoted the uptake of cystine. This led to an increase in intracellular cysteine levels, serving as a building block for protein synthesis and a substrate to avoid oxidative damage, ultimately contributing to cell proliferation. These observations indicate that CD44-positive hepatocytes are highly proliferative. Given that CD44 expression decreases significantly during hepatocyte maturation, CD44 seems to help maintain small hepatocytes and HPCs in a proliferative and undifferentiated state. These findings suggest that CD44 is a key regulator of cell proliferation and maturation, which are essential for recovering functional hepatocytes during liver regeneration (Figure 2). However, there are currently no studies elucidating the mechanisms through which CD44 affects proliferation and differentiation in the liver.

Several studies have shown the involvement of CD44 in the proliferation of non-hepatic cells, which can provide insights into the role of CD44 in liver regeneration. CD44 has been reported to regulate cell proliferation and survival by modulating cell-cycle-related proteins and β-catenin signaling in fibroblasts and lymphoblasts [102,103]. In fibroblasts, treatment with an anti-CD44 antibody upregulated the expression of cyclin A and the activity of cyclin A-associated dependent kinase 2, and it promoted cell detachment and apoptosis [103]. CD44 was also found to regulate cyclin D1 in K562 cells, a lymphocyte cell line. Suppression of CD44 by shRNA in K562 cells resulted in G0/G1 arrest, decreased cyclin D1, increased p21, and significantly reduced proliferation [102]. In addition, CD44 enhances cell proliferation and survival by modulating β-catenin signaling. Reduced CD44 levels downregulated β-catenin and upregulated β-catenin phosphorylation, which led to the inactivation of β-catenin signaling in K562 cells. In hepatocytes, it is well known that stabilized β-catenin translocates into the nucleus and activates genes that induce cell proliferation, promoting hepatocyte proliferation [104,105,106]. Hence, it is plausible that CD44-expressing hepatocytes are proliferative because CD44 regulates cell-cycle-related proteins and activates β-catenin to induce proliferation.

CD44 has been reported to impact cell differentiation. Isono et al. [107] reported that multipotent human primary synovial fibroblasts with high levels of CD44 expressed stem cell markers such as CD73, CD90, and CD105, while synovial fibroblasts with low levels of CD44 rarely had these markers. CD44 also regulates hematopoietic cell differentiation [108,109]. CD34-positive hematopoietic stem cells isolated from the thymus express CD44, but its levels decline during differentiation into thymocytes [108]. T cells differentiate into distinct effector subtypes such as T helper (Th)1 or Th2 depending on the presence of CD44 [109]. CD44 polarized CD4-positive T cells into Th1 cells, not Th2 cells. CD44-depleted mice showed dominant Th2 cytokines with diminished Th1 cytokines in the serum cytokine profile. The observed differences in the differentiation of Th cells have been reported to be caused by the regulatory effect of CD44 at the transcription level. CD44ICD interacted with the janus kinase/signal transducers and activators of transcription signaling pathway and induced the expression of T-box protein 21 and GATA binding protein 3, key transcription factors controlling gene expression, which selectively allowed Th cells to differentiate into Th1 or Th2.

The effects of CD44 on the proliferation and differentiation of non-hepatic cells may provide mechanistic insights into its impact on cell proliferation and maturation in the liver. However, these cellular responses mediated by CD44 may vary depending on the cell type. Therefore, further studies are needed to elucidate the functions of CD44 in liver cell proliferation and differentiation.

## 5. Conclusions

CD44 is universally expressed in various cells and tissues [19,20,21,25,26,27,28]. CD44 expression in the liver is not as high as in other organs, such as the skin, lungs, and intestines [19,29]. Because of the relatively low levels of CD44 in the normal liver, its expression can be significantly altered in response to liver damage, influencing pathogenic reactions, including dysregulated metabolism, proliferation, inflammation, and fibrosis. These pathophysiological effects of CD44 in the liver suggest that CD44 could be a promising therapeutic target for the treatment of liver disease. However, to date, research on CD44 in the liver has primarily focused on cancer, and not much is known about the role of CD44 in other disease conditions. Considering the differences between cancer and non-cancer diseases and the significant influence of CD44 in the liver, additional studies are required to elucidate the contribution of CD44 to the pathogenesis of liver disease. CD44 contributes to disease progression, while also inducing the proliferation and differentiation of hepatocytes necessary for liver regeneration. Its actions differ depending on the interacting partners and the types of cells expressing CD44. While existing research has mainly focused on the ligands that interact with CD44, it is important to understand the intracellular molecules activated by CD44 and the interaction partners, translocation, and activation processes of CD44 within cells. Hence, further in-depth investigations are necessary to determine the cell-specific roles of CD44 and its regulatory mechanisms in cellular responses and liver regeneration.

Although the role of CD44 in the liver remains unclear, recent studies indicate that CD44 has the potential to act as a key regulator in liver regeneration and disease progression. Therefore, gaining an understanding of the sophisticated mechanisms of CD44 may help in the development of treatments that selectively promote liver regeneration while suppressing the pathogenesis of liver disease.

## Figures and Tables

**Figure 1 ijms-25-04749-f001:**
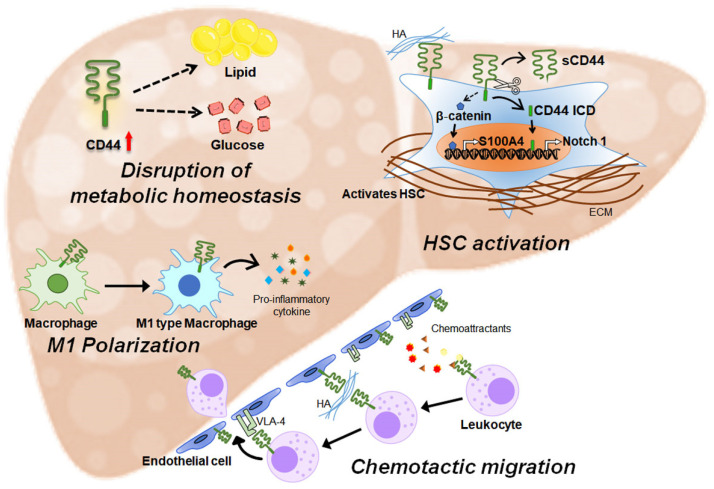
A schematic diagram depicting the actions of CD44 in the liver’s pathogenesis. Hepatic expression of CD44 is upregulated in liver diseases such as hepatic steatosis, metabolic dysfunction-associated steatohepatitis, HBV or HCV infection, and congestive hepatopathy with fibrosis. Upregulated CD44 disturbs metabolic homeostasis by impacting glucose and lipid metabolism in the liver, but the mechanisms underlying its effect have not been fully revealed. CD44 contributes to polarization of hepatic macrophages into M1 macrophages, which release pro-inflammatory cytokines. CD44 also promotes chemotactic migration of leukocytes to infiltrate into the injured liver. CD44 on leukocytes recognizes chemoattractants, rolls toward endothelial cells, and binds to HA presented by endothelial cells, tethering to endothelial cells. Then, CD44 carries out high-affinity binding with α4β1 integrin (VLA-4) of endothelial cells, stabilizing leukocytes’ adhesion and facilitating extravasation to damaged areas. In hepatic stellate cells (HSCs), the main fibrogenic cells, CD44 promotes HSC activation. CD44 interacting with HA activates β-catenin, which directly binds to the S100A4 promoter to induce its expression, leading to HSC activation. Proteolytic cleavage of CD44 has been reported to occur during HSC activation, and soluble CD44 (sCD44) and the CD44 intracellular domain (CD44ICD) are elevated. Released CD44ICD translocates into the nucleus and binds to the promoter/enhancer region of Notch1, inducing its transcription. Dashed lines indicate pathways that have not yet been fully elucidated.

**Figure 2 ijms-25-04749-f002:**
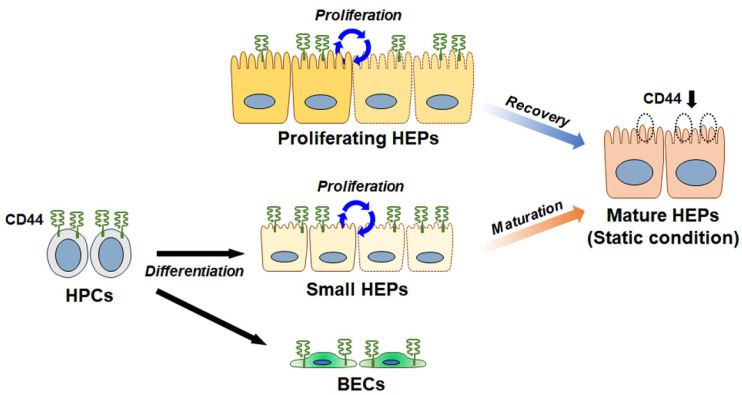
A simplified diagram of expressional changes in CD44 during proliferation and maturation of liver cells. CD44 is highly expressed in hepatic progenitor cells (HPCs) that have potential for differentiation into both hepatocytes and biliary endothelial cells (BECs). Small hepatocytes (HEPs) differentiated from HPCs also express CD44 on their surface and have both proliferative and hepatic characteristics. However, CD44 on small HEPs rapidly decreases during maturation of small HEPs into mature HEPs. Although mature HEPs barely express CD44 and are mostly static in the cell cycle, proliferating HEPs contain significantly higher levels of CD44 than mature HEPs, providing functional mature HEPs. CD44 seems to be involved in maintaining the proliferation and undifferentiation status of these cells. However, the mechanism through which CD44 affects proliferation and maturation in the liver is not elucidated.

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
