# Peer review of "Current Evidence and Perspectives of Cluster of Differentiation 44 in the Liver’s Physiology and Pathology"

_ijms, 2024, doi:10.3390/ijms25094749_

Round 1
Reviewer 1 Report
Comments and Suggestions for Authors
The review is well written and informational. I do not have any concern in publishing the manuscript as it is.
The current review article majorly addressed the role of CD44 in liver diseases including lipid and glucose dysregulation, inflammation, fibrosis, which are known risk factors for MASLD. In addition, the article also shed light on its role in liver regeneration such as hepatocyte proliferation under physiological and pathological conditions.
Narrating CD44 role in liver diseases such as lipid and glucose dysregulation, inflammation and fibrosis are original. Moreover, describing CD44 role in liver regeneration including proliferation and differentiation are relevant to the filed. Several review articles provided information on the role of CD44 in liver cancer, particularly in the initiation and progression of liver cancer. However, there are very few or no study provided complete information on the role of CD44 in early stages of liver cancer such as lipid dysregulation, inflammation, and fibrosis, which are considered as major contributing factors for liver cancer. The authors support their conclusion with appropriate references.
In the CD44 is involved in liver metabolism section, the details included in the discussion suggests that CD44 in the hepatocytes regulate lipid/glucose metabolism. However, the studies cited use CD44-/- mice and those studies may not delineate the hepatocyte-specific role of CD44 in metabolism. The authors need to clarify it.
In the figure 1 it would be good to provide more information like under what conditions CD44 such as HFD diet, fatty liver etc. levels increases and the crosstalk between the CD44 mediated lipid dysregulation, inflammation and fibrosis along with its independent roles. Please refer to https://www.sciencedirect.com/science/article/pii/S016882781730137X.
Author Response
The review is well written and informational. I do not have any concern in publishing the manuscript as it is.
The current review article majorly addressed the role of CD44 in liver diseases including lipid and glucose dysregulation, inflammation, fibrosis, which are known risk factors for MASLD. In addition, the article also shed light on its role in liver regeneration such as hepatocyte proliferation under physiological and pathological conditions.
Narrating CD44 role in liver diseases such as lipid and glucose dysregulation, inflammation and fibrosis are original. Moreover, describing CD44 role in liver regeneration including proliferation and differentiation are relevant to the filed. Several review articles provided information on the role of CD44 in liver cancer, particularly in the initiation and progression of liver cancer. However, there are very few or no study provided complete information on the role of CD44 in early stages of liver cancer such as lipid dysregulation, inflammation, and fibrosis, which are considered as major contributing factors for liver cancer. The authors support their conclusion with appropriate references.
In the CD44 is involved in liver metabolism section, the details included in the discussion suggests that CD44 in the hepatocytes regulate lipid/glucose metabolism. However, the studies cited use CD44-/- mice and those studies may not delineate the hepatocyte-specific role of CD44 in metabolism. The authors need to clarify it.
: As you explained, the cited articles [J Hepatol. 2017 Aug;67(2):328-338.; PLoS One. 2013;8(3):e58417.; Diabetes. 2015 Mar;64(3):867-75.] showed that CD44 deficiency influenced lipid and glucose metabolism in the liver of HFD-fed mice, but did not unveil whether the metabolic changes mediated by CD44 were specific to hepatocytes. Hence, we neither described nor suggest the hepatocyte-specific role of CD44 in metabolism. We just explained that CD44-deficiecy reduces metabolic dysfunction such as glucose tolerance and insulin sensitivity, and decreases expression of lipid metabolism-related genes including CD36, Elovl-5, and Elovl-7 in the livers of mice, not in specifically stating in hepatocytes. As you know, the detailed mechanism underlying the regulatory action of CD44 in lipid and glucose metabolic process in hepatocytes has not been fully elucidated, although hepatocytes are primarily responsible for lipid and glucose metabolism in the liver. Therefore, we described, “These findings indicate that CD44 influences glucose and lipid metabolism in the liver, but the exact mechanism is poorly understood due to limited research.” in the manuscript.
In the figure 1 it would be good to provide more information like under what conditions CD44 such as HFD diet, fatty liver etc. levels increases and the crosstalk between the CD44 mediated lipid dysregulation, inflammation and fibrosis along with its independent roles. Please refer to https://www.sciencedirect.com/science/article/pii/S016882781730137X.
: It has been reported that CD44 expression is elevated in various liver disease, but it is poorly understood how increased CD44 impacts the pathological features in liver disease, excluding cancer. Also, it remains unclear what increases CD44 level in the liver. In addition, we looked at the article which you referred and CD44 involvement in immune response in the liver was depicted. The figure in our manuscript illustrated the involvement of CD44 in the regulation of HSC activation and lipid & glucose metabolism in the liver in addition to the regulation of inflammation. Given the unclear mechanism of CD44 action in these responses in the liver, it is difficult to reflect the information which you asked into the figure. Hence, we marked the possible contribution (unclear mechanisms) of CD44 to these responses in the liver with dashed lines, as described in the figure legend. Known mechanisms are indicated in detail in the figure, such as mechanism explaining CD44 action in inducing HSC activation and liver fibrosis. Also, we depicted steatohepatitis with fibrosis in the background of Figure 1 because the hepatic role of CD44 has been mostly studied in steatohepatitis models induced by HFD, MCD and CDAA diet. Please understand that we provided information in the best way possible. We added the disease condition in which CD44 is upregulated in the legend of Figure 1: “Hepatic expression of CD44 is upregulated in liver disease such as hepatic steatosis, metabolic dysfunction-associated steatohepatitis, infection of HBV or HCV, and congestive hepatopathy with fibrosis.
Reviewer 2 Report
Comments and Suggestions for Authors
In the manuscript entitled “Current evidence and perspectives of cluster of differentiation 44 in the liver physiology and pathology”, Jinsol Han et al. reviewed the general information of CD44 and the function of CD44 in the pathophysiology of the liver. Authors mentioned that CD44 is associated with pathogenesis by impacting cellular response such as metabolism, proliferation, differentiation, and activation. Authors also suggested the need for further research on CD44. The manuscript is well written and the topic is very meaningful.
Author Response
In the manuscript entitled “Current evidence and perspectives of cluster of differentiation 44 in the liver physiology and pathology”, Jinsol Han et al. reviewed the general information of CD44 and the function of CD44 in the pathophysiology of the liver. Authors mentioned that CD44 is associated with pathogenesis by impacting cellular response such as metabolism, proliferation, differentiation, and activation. Authors also suggested the need for further research on CD44. The manuscript is well written and the topic is very meaningful.
: Thank you for your time and effort in reviewing our manuscript.
Reviewer 3 Report
Comments and Suggestions for Authors
The paper by Dr. Jinsol Han reviews CD44 function in liver disease except cancer. CD44 is ubiquitously expressed and regulate metabolism, proliferation, differentiation, and activation, in different cells. Many studies have reported CD44 is related to liver disease such as hepatic steatosis, fibrosis and HCV infection. However, the mechanisms remain unknown. This review contributes to understand CD44 function each cell type in liver but some of the methods need to be clarified.
1. CD44 is a major activation and memory marker of T cells. This review lacks interaction CD44 expressed T cells and liver physiology and pathology. For example, Guidotti et al. pointed out that circulating CD8 T cells arrest within liver sinusoids by docking onto platelets previously adhered to sinusoidal hyaluronan via CD44. Please clarify the role of CD44 expressed T cells in liver.
Author Response
The paper by Dr. Jinsol Han reviews CD44 function in liver disease except cancer. CD44 is ubiquitously expressed and regulate metabolism, proliferation, differentiation, and activation, in different cells. Many studies have reported CD44 is related liver disease such as hepatic steatosis, fibrosis and HCV infection. However, the mechanisms remain unknown. This review contributes to understand CD44 function each cell type in liver but some of the methods need to be clarified.
- CD44 is major activation and memory marker of T cells. This review lacks interaction CD44 expressed T cells and liver physiology and pathology. For example, Guidotti et al. pointed out that circulating CD8 T cells arrest within liver sinusoids by docking onto platelets previously adhered to sinusoidal hyaluronan via CD44. Please clarify the role of CD44 expressed T cells in liver.
: Thank you for your comment. As you suggested, we added the description of T cells-related CD44 in liver disease by referring to several papers including the one you recommended in the revised manuscript, 3.2 CD44 regulates inflammation: “CD44 is also involved in activation and recruitment of T cells. CD44 expression is upregulated during T cell activation, and CD44-positive activated T cells are highly abundant in the livers of mice injured by hepatitis virus infection, HFD or ischemia/reperfusion, and is correlated with disease progression [Eur J Immunol. 2007 Apr;37(4):925-34.; Front Pharmacol. 2019 Mar 13:10:244.; Cell. 2015 Apr 23;161(3):486-500.; FASEB J. 2017 Nov;31(11):4796-4808.; J Leukoc Biol. 2005 Aug;78(2):412-25.]. CD44 mediates conjugation between T cells and antigen-presenting dendritic cells (DCs) to promote T cell activation [Commun Integr Biol. 2010 Nov;3(6):508-12.; J Immunother. 2004 Jan-Feb;27(1):1-12.; J Leukoc Biol. 2008 Jul;84(1):134-42.]. Funken et al. reported that blocking CD44 by monoclonal antibody (mAb) interrupted interaction between DCs and T cells, and induced T cell apoptosis, inhibiting immune response in the post-ischemic liver [FASEB J. 2017 Nov;31(11):4796-4808.]. CD44 interacts with hyaluronan expressed on liver sinusoidal endothelial cells, and leads to T cell recruitment into the inflamed liver sinusoids [J Exp Med. 2008 Apr 14;205(4):915-27.; Front Immunol. 2015 Feb 17:6:68.]. Treatment of anti-CD44 mAb decreased T cell accumulation in the injured liver along with reduced T cell motility [FASEB J. 2017 Nov;31(11):4796-4808.; Gastroenterology. 2006 Feb;130(2):482-92.]. CD44 on platelets is reported to be implicated in recruitment of T cells in the HBV-infected liver [Cell. 2015 Apr 23;161(3):486-500.]. Circulating T cells are arrested by docking to CD44 expressed in platelets, and these T cells crawl along liver sinusoids to search hepatocellular antigen.”
Round 2
Reviewer 3 Report
Comments and Suggestions for Authors
The review is written better than last time. I have no comments.